# Selective Passivation of Three-Dimensional Carbon Microelectrodes by Polydopamine Electrodeposition and Local Laser Ablation

**DOI:** 10.3390/mi13030371

**Published:** 2022-02-26

**Authors:** Babak Rezaei, Saloua Saghir, Jesper Yue Pan, Rasmus Schmidt Davidsen, Stephan Sylvest Keller

**Affiliations:** National Centre for Nano Fabrication and Characterization, DTU Nanolab, Technical University of Denmark, 2800 Konges Lyngby, Denmark; saghir.saloua@gmail.com (S.S.); jesyup@dtu.dk (J.Y.P.); rasda@dtu.dk (R.S.D.); suke@dtu.dk (S.S.K.)

**Keywords:** electrodeposition, 3D pyrolytic carbon microelectrodes, pyrolysis, photolithography, polydopamine, laser ablation

## Abstract

In this article, a novel approach for selective passivation of three-dimensional pyrolytic carbon microelectrodes via a facile electrochemical polymerization of a non-conductive polymer (polydopamine, PDA) onto the surface of carbon electrodes, followed by a selective laser ablation is elaborated. The 3D carbon electrodes consisting of 284 micropillars on a circular 2D carbon base layer were fabricated by pyrolysis of lithographically patterned negative photoresist SU-8. As a second step, dopamine was electropolymerized onto the electrode by cyclic voltammetry (CV) to provide an insulating layer at its surface. The CV parameters, such as the scan rate and the number of cycles, were investigated and optimized to achieve a reliable and uniform non-conductive coating on the surface of the 3D pyrolytic carbon electrode. Finally, the polydopamine was selectively removed only from the tips of the pillars, by using localized laser ablation. The selectively passivated electrodes were characterized by scanning electron microscopy, cyclic voltammetry and electrochemical impedance spectroscopy methods. Due to the surface being composed of highly biocompatible materials, such as pyrolytic carbon and polydopamine, these 3D electrodes are particularly suited for biological application, such as electrochemical monitoring of cells or retinal implants, where highly localized electrical stimulation of nerve cells is beneficial.

## 1. Introduction

Electrical stimulation of cells and tissue is an important part of a broad range of prosthetic and diagnostic research, such as neural stimulation targeting, e.g., brain disorders [1], cochlear [2,3] and retinal diseases [4]. The distance between stimulating electrode and target cell is crucial, since the electric field from the electrode decreases quadratically with the distance, while the stimulation voltage and current are often limited by prosthetic design (e.g., in case of photovoltaic stimulation), the water window and concerns of tissue heating. Specifically, in the case of retinal prosthesis [5,6,7,8], miniaturization of pixels with the aim of higher spatial resolution leads to a lower penetration depth of the electric field into the tissue. Thus, there is a need for approaches that bring the stimulation electrodes closer to the target cells. A promising approach in this regard is to fabricate 3D electrodes that utilize the tendency of cells to migrate into voids and spaces between the electrodes. Palanker et al. [9] showed that retinal tissue migrated towards and around 65 μm high pillars six weeks after implantation in rats, compared to a 40 μm separation between tissue and the flat implant control. While those pillars were made from electrically inactive material in order to focus on the cell migration, electrically active pillar electrodes with similar dimensions have been demonstrated elsewhere. Ho et al. [10] made ‘mushroom’ shaped Si pillars with sputtered IrO_x_ capping and showed promising stimulation capabilities measured by visually evoked potentials and grating acuity. Davidsen et al. [11] showed preliminary electrical data such as charge storage capacity of pyrolytic carbon pillars aimed for subretinal prosthesis.

A challenging aspect of employing pillar electrodes in a stimulating device is the confinement of the electric field to the desired target site. Specifically, the pillars tend to be made from a single, conductive material or a semiconductor coated with a suitable conductor such as IrO_x_ [12]. In both cases the electric field will emit from the entire pillar surface, including the vertical sidewalls. This poses a very likely risk of cross-talk between neighbouring pillar electrodes and loss of charge to the medium surrounding the target cell to be stimulated. Ideally, current and electric field should be confined to the intended site of stimulation, which is often the top of the pillar facing the tissue. In order to realize such a confinement, the pillar sidewalls must be passivated with a biocompatible insulator, while passivation at the very top of each pillar should be prevented. However, this type of selective passivation of 3D microelectrodes is challenging to implement at the micron scale.

Dopamine is a catecholamine neurotransmitter, which is an aromatic and redox-active compound that can be polymerized spontaneously in aerated alkaline conditions [13,14]. Inspired by marine mussels and their incredible adhesive properties, Messersmith et al. showed that immersion in an aqueous tris buffer saline (TBS) solution at pH 8.5 with 2 mg mL^−1^ of dopamine for 24 h resulted in a 50 nm polydopamine (PDA) coating [13]. Due to the existence of the residual catechol and alkylamine functionalities within the PDA structure, self-polymerized PDA creates a tight adhesive coating over a wide variety of substrates, including noble metals, carbon oxides, polymers, semiconductors, and ceramics with complex shapes [15,16,17]. Recently, PDA has been used as an organic coating for a variety of applications in different fields, ranging from enzyme immobilization to wastewater purification [18,19]. However, it is difficult or almost impossible to control the PDA film homogeneity, and tailor the coating thickness and properties through the self-polymerization of dopamine [16,20]. A promising, but less explored approach to modulate the polymerization process of dopamine is the use of electrochemical polymerization. There, the dopamine polymerization is initiated by applying a potential to a conductive material that is sufficient to oxidize or reduce the monomers. Considering the accuracy of electrochemical methods for precise control of oxidation and reduction reactions, the electropolymerization of dopamine under neutral conditions provides an alternative approach for tailoring the PDA coating thickness and properties [14]. Wang et al. have described an efficient method to polymerize dopamine in a controlled way on a gold-coated silicon wafer using cyclic voltammetry [14]. Dopamine electropolymerization provides high deposition rates [14], excellent uniformity of the coating, reproducibility, eco-friendliness and mechanical stability for in vivo testing [21]. With the electrodeposition method, PDA layers between 10 nm to 1 μm thickness [14,22,23] have been achieved.

Pyrolytic carbon electrodes have been intensively explored for bioelectrochemistry due to their advantageous properties, including chemical inertness, low fabrication costs, wide electrochemical potential window and electrocatalytic activity for many redox reactions [24]. The pyrolysis of photoresists patterned by lithography permits one to obtain electrodes with tailored designs and well-defined geometries in a highly reproducible way [25,26]. The subsequent pyrolytic carbon has outstanding properties, such as chemical stability, strength, fatigue and wear resistance, lubricating properties, stress resistance, and excellent biocompatibility. Microfabricated pyrolytic carbon (PyrC) electrodes are already used for countless applications such as electrocatalysis [27], energy conversion and storage [28,29,30], electroanalysis [31], sensors and biosensors [32,33], cell culturing and differentiation [34,35].

PyrC is composed of amorphous and graphitic micro-domains, which make it a good electrical and thermal conductor, whereas polydopamine films are electrically insulating. Both materials are highly biocompatible [15,36,37] and particularly suited for surface functionalization [20,38]. E. Peltola et al. have shown that PDA had stronger adsorption on PyrC electrodes compared to other carbon-based surfaces [39]. Applications of PDA films on PyrC electrodes include pH sensing in unbuffered solutions [40], microbial fuel cells [41], green electrocatalysis [42], energy conversion and storage, cellular and molecular immobilisation [43]. However, in all those studies, PDA coatings conformally cover the complete electrode. The selective removal of PDA from the PyrC surface at some well-defined locations would allow for a local difference of electrical conductivity and chemical reactivity. This difference in physicochemical properties is particularly interesting for selective immobilization of biological entities, such as enzymes, proteins, cells and molecules. 3D electrodes with selective PDA removal at the tips of the pillars are propitious inducing electrical impulses limited to a small spot size.

In this paper, we present a novel method for fabrication and selective passivation of 3D microelectrodes based on PyrC micropillars. The electrodes were fabricated using UV photolithography with the epoxy-based photoresist SU-8 followed by pyrolysis under inert atmosphere. Electropolymerization of dopamine on the carbon surface was optimized to ensure complete passivation of the 3D electrodes. This was followed by implementation of laser ablation to locally and selectively remove PDA from the carbon and to provide access for electrochemical reactions at well-defined locations. The surface of the electrode was characterized at each stage of the processes using scanning electron microscopy (SEM), electrochemical impedance spectroscopy (EIS) and cyclic voltammetry (CV).

## 2. Materials and Methods

### 2.1. Materials

SU-8 2035 and 2075 were obtained from MicroChem, (Newton, MA, USA). Dopamine hydrochloride was purchased from Merck (Darmstadt, Germany). Potassium hexacyanoferrate (III) and potassium hexacyanoferrate (II) trihydrate were purchased from Fisher (Houston, TX, USA). The tris-buffered saline (TBS) was prepared from NaCl, KCl (both obtained from Sigma Aldrich, Deisenhofen, Germany) and Tris base purchased from Fisher. The phosphate buffer solution (PBS, pH 7.4) was obtained from Sigma Aldrich. All the chemicals were of analytical grade and deionized water was used throughout the experiments.

### 2.2. Fabrication of the 2D/3D Carbon Electrodes

The 3D electrodes were composed of 284 micropillars uniformly placed on a 2D circular base layer with a diameter of 4 mm. The overall fabrication process of the 2D and 3D electrodes is schematized in Figure 1 and has been described previously in detail [44]. Briefly, 6-inch silicon wafers with 600 nm Si oxide grown by wet oxidation were used as substrates. The wafers were then dehydrated in an oven at 250 °C for 30 min before being coated with 17 μm negative photoresist (SU-8 2035 from MicroChem, USA) (Figure 1A). The SU-8 layer was patterned by UV exposure in soft-contact mode performed two times at 250 mJ cm^−2^ with a chromium mask to delimit the area of the circular 2D electrode followed by a post exposure bake (Figure 1B). To fabricate SU-8 pillars with a nominal diameter of 100 μm, two additional spin-coating steps were performed to deposit approximately 500 μm of SU-8 2075 (Figure 1C). The second UV photolithography process occurred with 4 × 250 mJ cm^−2^ followed by a post-exposure bake and development in propylene glycol methyl ether acetate (PGMEA) (Figure 1D,E). A flood exposure at 2 × 250 mJ cm^−2^ combined with a hard bake at 90 °C for 15 h allowed to complete the crosslinking of the photoresist.

To convert the SU-8 photoresist into pyrolytic carbon, a pyrolysis step was performed in a PEO604 furnace (ATV Tech., Germany) at 1100 °C for 5 h with a temperature ramp of 10 °C min^−1^ under nitrogen flow (Figure 1F).

For connection of the pyrolytic carbon electrodes to the external measurement setup during electrochemical characterization and PDA electrodeposition, 200 nm thin platinum leads and a contact pad were deposited via e-beam evaporation (Alcatel SCM 600) (Figure 1G) using a shadow mask previously shaped by laser machining. Finally, to passivate the Pt leads and precisely define the WE area, a 5 μm thick film of SU-8 2005 was deposited by spin coating at 2000 rpm for 30 s with an acceleration of 500 rpm s^−1^ followed by solvent evaporation for 2 h at room temperature (Figure 1H). Similar to the previous photolithography steps, the wafers underwent a first UV exposure at 2 × 250 mJ cm^−2^, a post-exposure bake at 50 °C for 2 h, a development which consisted in immersing the wafers two times in PGMEA for 5 min and a rinsing in isopropanol and drying in air. This was followed by another flood exposure at 2 × 250 mJ cm^−2^ with a waiting time of 30 s and a hard bake at 90 °C for 15 h. The fabrication process was concluded by the dicing of the wafer into individual electrode chips using a laser micromachining tool (microSTRUCT, 3D Micromac AG, Germany). For fabrication of the 2D electrodes, the process was identical except for the deposition and patterning of the thick SU-8 layer (Figure 1C–E). Photographs of 2D and 3D electrodes are presented in Appendix A.

### 2.3. Electropolymerization of Dopamine

The experiments were conducted in a custom-made 3D printed electrochemical cell with a three-electrode arrangement and a nitrogen gas inlet tube (Appendix A). The 3D carbon microelectrode (3DCME) was used as a working electrode (WE), with a silver/silver chloride (Ag/AgCl in 1 M KCl) reference electrode (RE), and a platinum wire as counter electrode (CE). The electrochemical characteristics of the carbon electrodes before and after surface modifications were evaluated by cyclic voltammetry (CV) and electrochemical impedance spectroscopy (EIS) techniques, using a PalmSens4 potentiostat with PSTrace software (PalmSens BV, The Netherlands). The electropolymerization process of PDA on the surface of the 3DCMEs was done potentiodynamically in the range from −0.2 to 0.8 V with different potential scan rates of 10, 20 and 50 mV s^−1^ and variable cycle numbers. The WE was first rinsed with deionized water and isopropanol and dried by blowing with pure compressed nitrogen gas. The electrodes then underwent an oxygen plasma treatment (at 120 W and 0.6 mbar for 75 s) to remove eventual residual organic contaminations and increase the hydrophilicity of the carbon electrode surface [45,46]. The electropolymerization process was carried out in 4 mL TBS solution (25 mM Tris, 140 mM NaCl, and 3 mM KCl at pH 7.4) which was bubbled with pure nitrogen gas for 10 min prior to insertion of 1 mg mL^−1^ dopamine hydroxide. Deoxygenation of the precursor solution was performed with pure nitrogen. The electrode was then rinsed by dipping several times in DI water and left to dry. Finally, the samples underwent a thermal treatment under an inert atmosphere (200 sccm of nitrogen) in a ceramic tube furnace (GSL-1700x, MTI Corporation) at 100 °C for 1 h, to stabilize the electrodeposited polymer layer on the pyrolytic carbon surface.

### 2.4. Electrochemical Characterization

CV and EIS measurements were carried out in a solution of phosphate buffered saline (PBS 10 mM, pH 7.4) containing 10 mM [Fe(CN)_6_]^−3/−4^ redox probe as the electrolyte solution. For CV measurements, a potential range of −0.6 to 0.8 V vs. RE was applied to the WE at a scan rate of 50 mV s^−1^. The EIS spectra were recorded in a frequency range from 0.01 Hz to 100 kHz with an amplitude of 10 mV, at a constant potential of 0.3 V, and fitted using EIS analysis software. Both CV and EIS experiments were repeated at least three times (*n* ≥ 3) for different electrodes.

### 2.5. Selective Removal of Electropolymerized PDA

A 50 ps UV laser on the microSTRUCT laser micromachining tool was used to selectively remove the electropolymerized dopamine at the tip of the micropillars on the 3D carbon electrode. The laser ablation process was performed with a power of 25 W at 50%, with a 200 kHz frequency and a wavelength of 355 nm. The focal distance was equal to 103 mm and the spot size was between 10 and 15 μm. By using the same pattern for laser ablation as the one used for fabrication of SU-8 micropillars by UV lithography, an alignment can be performed to able the PDA on all 284 pillars in one single process lasting only a few minutes. A two-point alignment was performed using the central pillar and one in the extremity, followed by a focused alignment before ablation. The laser ablation on each micropillar was performed writing 5 concentric circles with diameters of 5, 15, 25, 35 and 45 μm with a laser speed of 500 mm s^−1^.

## 3. Results and Discussion

### 3.1. Electrochemical Characterization of 2D/3D Electrodes

In order to electrochemically characterize the electrodes, CV measurements were performed on both 2D and 3D electrodes. In these experiments, the reversible character of the electron transfer reactions at the interface of the surface of the electrodes and the electrolyte was studied, with a scan rate varying from 20 to 200 mV s^−1^. For the [Fe(CN)_6_]^−3/−4^ redox probe, typically a linear correlation has been observed between the square root of the scan rate and the resulting peak current [27]. The Randles-Sevcik equation (Equation (1)) links the current to the scan rate, the concentration of the electrolyte and the electroactive surface area:(1)$$i_{p}=\left(2.69\times 10^{5}\right)n^{3/2}AD^{1/2}C\vartheta^{1/2}$$
where, *n* is the number of electrons involved in the reaction (*n* = 1 for [Fe(CN)_6_]^−3/−4^), *A* is the electroactive surface area (cm^2^), *D* is the diffusion coefficient of the redox probe in the solution (7.6 × 10^−6^ cm^2^ s^−1^ for [Fe(CN)_6_]^3−^), C corresponds to the concentration of the redox probe (mol cm^−3^) and ϑ is the potential scan rate (V s^−1^).

The 3D electrodes containing micropillars provided larger electroactive surface area than 2D electrodes leading to an increase in the redox peak currents observed in the cyclic voltammograms (Figure 2a). As an example, at 50 mVs^−1^ in 10 mM [Fe(CN)_6_]^3−/4−^, the oxidation peak currents are around 232 μA and 417 μA for the 2D and 3D electrodes, respectively. On average, an increase of 55.6% of the peak current was observed for the 3D microelectrodes compared to the 2D configuration. ImageJ software version 1.47t (National Institute of Health, Bethesda, MD, USA) was utilized to measure the diameter and height of the pillars from the top- and side-views SEM images, respectively. The 284 carbon micropillars have a diameter of 80 ± 1 μm and a height of ≈188 μm. In consequence, the geometric surface is 12.6 mm^2^ for the 2D electrode and 24.4 mm^2^ for the 3D electrode. Using the Randles-Sevcik equation (Equation (1)), the electroactive surface area (ESA) was estimated at 14.0 mm^2^ and 25.1 mm^2^ for the 2D and 3D electrodes, respectively. The ESA corresponds well to the geometric surface area, which explains the approximately two-fold increase in the peak currents for the 3D electrodes.

The anodic and cathodic peak currents increased with the scan rate for both 2D and 3D electrodes. Figure 2b shows the linear relationships of the redox peak currents versus the square root of scan rate. Increasing the potential scan rate leads to a continuous increase in the diffusion rate of electroactive species towards the electrode/electrolyte interface. Therefore, electroactive species from the bulk solution (zone affected by the working electrode) can migrate faster and in a higher number to the surface. This phenomenon leads to an increase in the peak current [47]. Similarly, a linear augmentation of the peak current was observed with increasing the concentration of the [Fe(CN)_6_]^3−/4−^species in the solution (results not shown). These observations indicate that the electron transfer process and electrochemical reactions on the surface of both 2D and 3D pyrolytic carbon electrodes are diffusion-controlled and reversible [48].

### 3.2. Electropolymerization of Dopamine

First, electropolymerization of dopamine on 3D pyrolytic carbon electrodes was investigated and optimized. A low concentration of dopamine of 1 mg mL^−1^ was utilized to minimize unwanted self-polymerization. This is very important because the PDA clusters formed through self-polymerization can be trapped within the electrodeposited PDA layer, thereby increasing the roughness of the coating layer and its heterogeneity. It is known that self-polymerization is induced by the oxidation of dopamine, which means that an oxygen-free environment is crucial [49]. Therefore, solution deoxygenation was performed all along the electropolymerization process using nitrogen bubbling at a neutral pH of 7.4, to provide a relatively efficient control of the undesirable self-polymerization of dopamine.

The CV curves of the electrochemical polymerization of dopamine at different scan rates of 50, 20 and 10 mV s^−1^ are presented in Figure 3a–c. Moreover, the values of second anodic peak currents as a function of the cycle number for the aforementioned potential scan rates are summarized in Figure 3d. The presence of several oxidation and reduction peaks in the cyclic voltammograms is due to the complex mechanism of electrochemical polymerization of dopamine with several oxidation and reduction states, where sub-products are being generated and transformed to other forms during the reaction. The mechanism of dopamine polymerization has been extensively investigated and explained in recent years [14,16,18,20,40,49,50,51].

Once the peak currents drop to almost zero, it means that the electrochemical reactions across the electrode/electrolyte interface are effectively prohibited, which can be attributed to the formation of a uniform thin layer of non-conductive PDA polymer on the surface of the 3D pyrolytic carbon electrode. Since a potentiodynamic (cyclic voltammetry) method was employed to polymerize dopamine and decorate the electrode surface with a PDA insulating layer, the kinetics of the electrochemical redox reactions (i.e., potential scan rate) was one of the most crucial parameters influencing the electrodeposition performance [16,52]. The scan rate corresponds to the speed at which the electrical potential is applied to the system and, therefore, indicates how fast the redox reactions occur at the surface of the working electrode. As summarized in Figure 3d, the second anodic peak current decreased and almost disappeared after the 15th cycle and 36th cycle at 10 and 20 mV s^−1^ scan rates, respectively, whereas it did not completely disappear even after 100 consecutive CV cycles at 50 mV s^−1^. At low scan rates (10 and 20 mV s^−1^), dopamine has sufficient time to undergo the reactions involved in its electropolymerization process [16]. After each cycle, a significant drop of the peak current was observed. This means that a larger fraction of the surface was covered with the PDA layer, resulting in a decreased area accessible for the electrochemical reactions between the surface of the electrode and the dopamine molecules within the solution. It can be seen from the CV results that the decay of the peak currents between each cycle for 10 mV s^−1^ scan rate was much larger than for 20 mV s^−1^. Overall, since the main goal of this study was to deposit a reliable and uniform non-conductive coating on the surface of the 3D pyrolytic carbon electrode, CV with 10 mV s^−1^ scan rate and a total number of 50 cycles was chosen as the optimum PDA electrodeposition condition.

Figure 4 displays SEM images of carbon micropillars before and after electropolymerization of dopamine with the optimized conditions. Before electropolymerization (Figure 4a–c), the micropillars had a smooth surface and uniform structure, which is known as the typical topography of SU-8-derived carbon materials [53,54,55]. The SEM images after the electropolymerization process (Figure 4d–f) confirm the formation of a uniform and recognizable PDA coating on the surface of the 3D carbon electrodes. It can be observed that the electrodeposition of PDA induces an increase in surface roughness and some small agglomerations, which could be attributed to the entrapment of self-polymerized PDA clusters into the electropolymerized PDA coating.

### 3.3. Electrochemical Characterization of PDA Coated 3D Carbon Electrodes

Next, the influence of PDA electrodeposition conditions on the passivation of the 3D pyrolytic carbon electrodes was investigated. For this purpose, the electrochemical activity of the electrodes coated with different scan rates was examined through CV measurements in a PBS solution (10 mM, pH 7.4) containing 10 mM [Fe(CN)_6_]^3−/4−^ as a reversible redox probe. Figure 5 shows the CV measurements of the electrodes that were coated for 50 consecutive potentiodynamic cycles at 50, 20 and 10 mV s^−1^ potential scan rates. For the electrode coated with the potential scan rate of 50 mV s^−1^, the final oxidation and reduction peak currents decreased in comparison with the bare electrode. However, the still very prominent redox peaks reveal that the electrode is not completely electrically insulated and that some areas are contributing to the electrochemical reactions. No electrochemical reactions of the [Fe(CN)_6_]^3−/4−^ redox probe were observed for the samples that were electropolymerized at 20 and 10 mV s^−1^ potential scan rates, meaning that the electrode surfaces were fully covered and passivated with an electrically insulating PDA coating. The electrodeposition of the PDA coating layer on the pyrolytic carbon electrodes also resulted in a visible colour change from a shiny black for the pyrolytic 3D carbon to a dark matte brown for the PDA coated electrodes [14,50]. It was observed that, after several days of storage in PBS solution, the detachment of the PDA coating layer could occur for electrodes coated with both 20 and 10 mV s^−1^ scan rates. Furthermore, a higher tendency for delamination was observed for the electrodes coated with 20 mV s^−1^ than the electrodes coated with 10 mV s^−1^ scan rate. To address this issue, a thermal relaxation process was introduced directly after electropolymerization of the PDA layer to reduce the residual stresses generated during the electrochemical polymerization process. As a result, no delamination effects were noticed even after three weeks of immersion in aqueous solution.

### 3.4. Selective Removal of PDA by Laser Ablation

During the local laser ablation process, the PDA on each micropillar was ablated using five concentric circles with diameters of 5, 15, 25, 35 and 45 μm starting from the center (Appendix A). In preliminary experiments, the effect of laser power was investigated and finally set at 50% of 25 W. Higher laser power induced damage of the 3D carbon electrodes, while lower laser power resulted in incomplete removal of the PDA layer on the top of the pillars. SEM images with different magnifications and viewing angles of the 3D carbon electrodes after selective removal of the PDA passivation layer from the tips of the micropillars are presented in Figure 6. As it can be seen from low magnification SEM images (Figure 6, bottom row), all the pillars were ablated successfully at their top, and the size and the location of the ablated areas were very consistent. The formation of concave structures on the tip of the micropillars (Figure 6, top row) could be ascribed to the utilized laser exposure pattern. For a circular laser pattern, the exposure dose is higher for the inner edge of the circle than for the outer one, which means that accumulated laser ablation towards the centre of the pillars should be more prominent that towards the edge. Moreover, a residual PDA coating layer displaying a significantly different roughness than the ablated areas can be distinguished at the edge of the micropillar tips. Since the size of the pattern designed for the laser ablation was slightly smaller than the diameter of the micropillars to prevent removal of the PDA layer coated on the sidewalls, the presence of some residual PDA domains at the tips of the pillars is inevitable. The same laser ablation experiment was done on multiple electrodes; very repeatable and reproducible results were observed.

### 3.5. Electrochemical Characterization after Local Laser Ablation

The effect of the laser ablation on the electrode passivation and electroactive area was again evaluated by CV with 10 mM [Fe(CN)_6_]^3−/4−^as a reversible redox probe. Figure 7a displays the CV measurements for the bare, PDA-coated and laser-ablated electrodes. The cyclic voltammogram of the laser-ablated electrode exhibited a pair of quasi-reversible redox peaks, which confirms the selective removal of the PDA passivating the tips of the pillars. The almost sigmoidal shape of the cyclic voltammograms after laser ablation could be attributed to appearance of radial diffusion profiles at the concave geometry of the micropillar tips. A similar ultra-microelectrode-like behaviour has earlier been observed for suspended pyrolytic carbon electrodes with feature sizes below 50 μm [25]. According to the Randles–Sevcik equation (Equation (1)) and considering the equivalent surface area of the tips of the pillars (284 micropillars with ~80 μm diameter), the peak current was theoretically expected to be around 25.4 μA. However, the CV measurements showed an anodic peak current of around 59.0 ± 4.9 μA, which corresponds to an increase in almost 40% for the measured peak current compared to the theoretical value. This augmentation might be due to the formation of concave structures on the tip of the micropillars, which could lead to a larger surface area for electrochemical redox reactions. In addition, the higher redox peak currents might also be due to local pyrolysis of the PDA layer in close proximity to the ablated area due to the very high temperature generated by the laser in a fraction of a second [56]. PDA is known as one of the precursors to fabricate super porous carbon materials, which can boost the electroactive surface area and, thereby, increase the redox peak currents measured for the electrodes after pyrolysis [57].

As a complementary method to CV, electrochemical impedance spectroscopy (EIS) was used to confirm the deposition of an electrically insulating layer onto the surface of the 3D carbon electrode, and then the selective removal of the PDA coating from the tips of the micropillars. EIS is an analytical technique that is very sensitive to processes taking place at the electrode/electrolyte interface and provides information such as the capacitance or the resistance of the double layer. Based on the recorded impedance spectra, the electrode and the electrolyte can be modelled with an equivalent electrical circuit allowing for subsequent extraction of specific values [58,59]. In the simplest case of a WE immersed in an aqueous solution containing a supporting electrolyte capable of undergoing redox reactions and assuming an electrochemical reaction limited by linear diffusion, the equivalent circuit describing most accurately the phenomenon occurring at the interface is a Randles circuit. It consists of a solution resistance R_s_ (corresponding to the electrolyte resistance), a constant phase element CPE in parallel with a charge transfer resistance R_ct_ and a Warburg impedance W, as shown in the inset of Figure 7b. The charge-transfer resistance is correlated with the electroactive area available for the transfer of charges between the redox probe in the solution and the surface of the electrode. In Figure 7b, the EIS spectra recorded for the 3D carbon electrodes are presented as a Nyquist plot, where the imaginary part of the impedance is plotted as a function of the real part of the impedance value. The radius of the semi-circle at high frequencies corresponds to the charge-transfer resistance and the slope of the linear part at lower frequencies is correlated with the diffusion coefficient. In the case of an insulating layer, no interaction of the probe species with the electrode should be observed, and the charge transfer resistance should be much higher than for a conductive electrode surface. The R_ct_ values of 1.46 ± 0.2 kΩ for the bare electrode, 321 ± 2.59 kΩ after the deposition of PDA and 18.86 ± 6.2 kΩ after laser ablation were derived from the equivalent circuit model. The significant increase in the R_ct_ after electropolymerization demonstrated that an electrically insulating layer was deposited. The extracted value for R_ct_ is similar to values reported earlier for PDA deposited on gold (R_ct_ = 93 kΩ) [14] and ITO (R_ct_ = 242 kΩ) [51]. However, direct quantitative comparison is difficult due to differences in electrode geometry, materials and electrolyte solutions used in these studies. The drastic decrease in R_ct_ after laser ablation confirmed the selective removal of the passivating PDA layer, allowing for redox reactions of the [Fe(CN)_6_]^3−/4−^ions limited to the top of the micropillar electrodes.

## 4. Conclusions

In this paper, we presented a novel method to fabricate and selectively passivate 3D pyrolytic carbon electrodes composed of micropillars with polydopamine. Conventionally, deposition of PDA coatings on substrate materials has been done by immersion in a solution of self polymerized dopamine. For conductive substrates, electropolymerization of dopamine with cyclic voltammetry should result in a more controlled coating process at higher deposition rates. In this sustainable process, no hazardous chemicals are required to induce polymerization. Here, the electrodeposition was optimized to achieve a conformal coating of 3D carbon microelectrodes with PDA completely passivating the electrode surface. As a second step, PDA was locally removed at the tips of the micropillars by laser ablation. This technique provided a high level of precision, notably by the very good control of the alignment of the laser, an excellent reproducibility, a possibility of automation to ablate an entire wafer and a fast process. This leads to a difference in physicochemical properties—such as electrical conductivity, chemical activity, cellular and molecular affinity—between the tips of the micropillars and the rest of the electrode. Removing the PDA layer from the tips of the micropillars and electrochemical activity of the pillars were approved by scanning electron microscopy, cyclic voltammetry and electrochemical impedance spectroscopy experiments, respectively. Due to the surface composed of highly biocompatible materials such as pyrolytic carbon and polydopamine, these 3D electrodes are particularly suited for biological application such as electrochemical monitoring of cells or retinal implants where highly localized electrical stimulation of nerve cells is beneficial.

## Figures and Tables

**Figure 1 micromachines-13-00371-f001:**
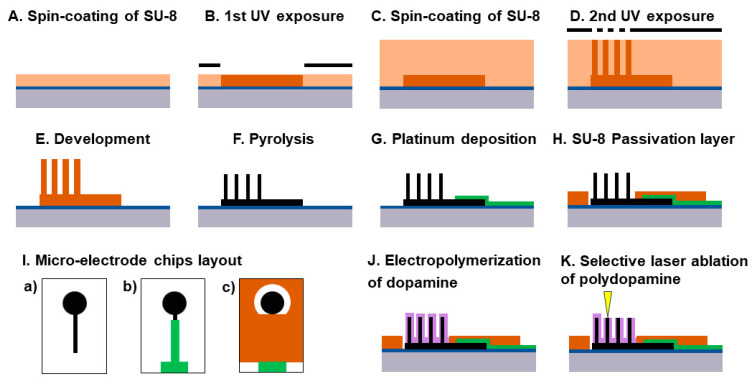
Schematic of carbon microelectrode fabrication: (**A**) SU-8 2035 is spin coated and soft baked; (**B**) 1st UV exposure and post exposure bake; (**C**) second layer SU-8 2075 is spin coated and soft baked; (**D**) 2nd UV exposure and post-exposure bake; (**E**) Development in PGMEA; (**F**) Pyrolysis under nitrogen at 1100 °C for 5 h; (**G**) e-beam deposition of Pt using a shadow mask; (**H**) SU-8 2005 passivation layer is spin coated and patterned by UV photolithograpy; (**I**) Top view of the microelectrode chip: (**a**) after SU-8 pyrolysis, (**b**) after Pt deposition and (**c**) after SU-8 passivation layer; (**J**) Electropolymerization of dopamine by CV; (**K**) Selective removal of polydopamine by laser ablation.

**Figure 2 micromachines-13-00371-f002:**
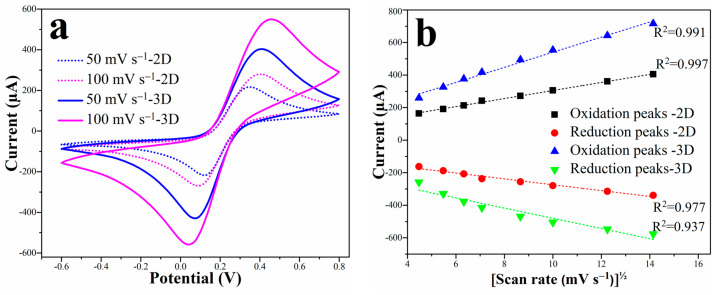
(**a**) Cyclic voltammograms for 2D (dashed lines) and 3D electrode at 50 mVs^−1^ (in blue) and 100 mV s^−1^ (in pink) in 10 mM [Fe(CN)_6_]^3−/4−^ solution prepared in 10 mM PBS (pH 7.4); (**b**) Peak currents versus square root of the scan rate.

**Figure 3 micromachines-13-00371-f003:**
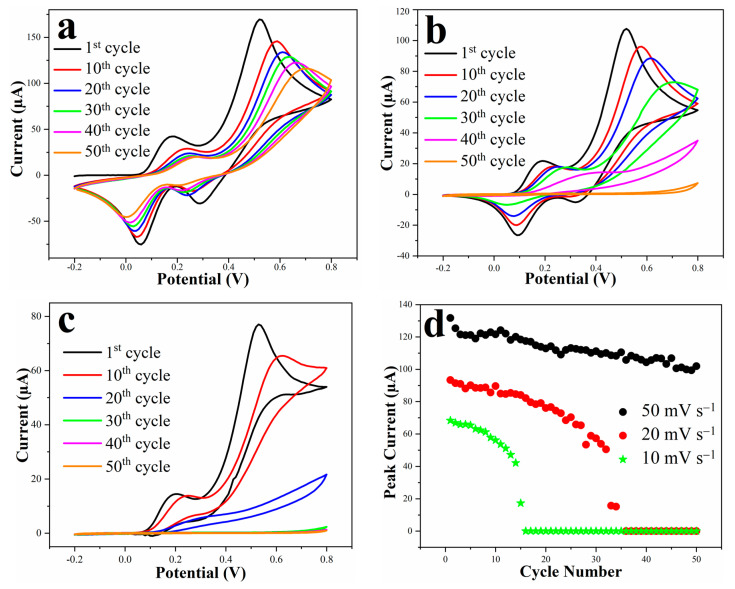
Monitoring cyclic voltammograms during the electropolymerization at: (**a**) 50 mV s^−1^; (**b**) 20 mV s^−1^; (**c**) 10 mV s^−1^; (**d**) Plot of the second anodic peak current of the CV of the electropolymerization of dopamine at different scan rates as a function of the number of cycles.

**Figure 4 micromachines-13-00371-f004:**
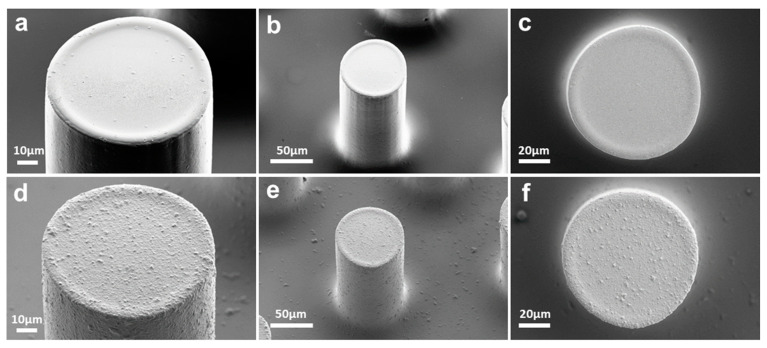
(**a**–**c**) SEM images of the bare 3D electrode; (**d**–**f**) SEM images of the 3D electrode coated with polydopamine at 10 mV s^−1^ for 50 cycles.

**Figure 5 micromachines-13-00371-f005:**
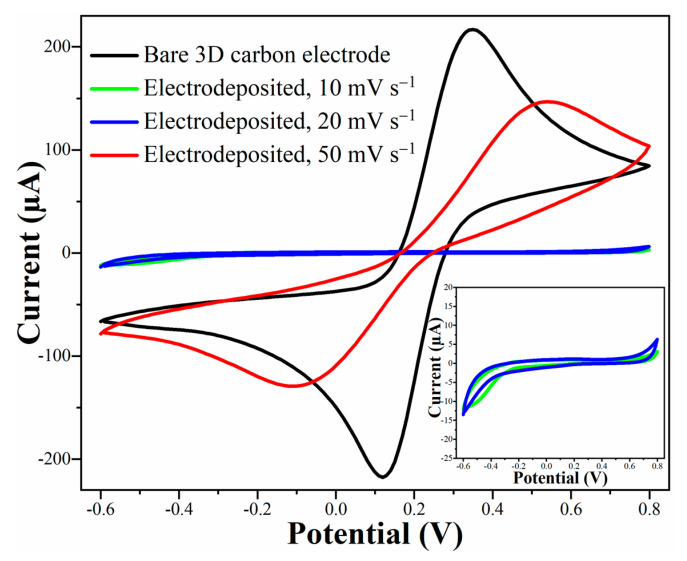
Cyclic voltammograms of the 3D pyrolytic carbon electrodes after electropolymerization of PDA for 50 consecutive potentiodynamic cycles at different scan rates of 50, 20 and 10 mV s^−1^. The CVs were obtained in 10 mM K_3_[Fe(CN)_6_]/K_4_[Fe(CN)_6_] solution prepared in 10 mM PBS (pH 7.4) at 50 mV s^−1^ scan rate.

**Figure 6 micromachines-13-00371-f006:**
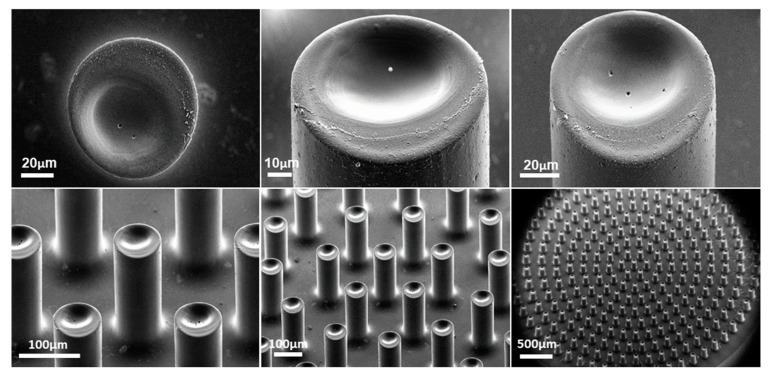
SEM images with different magnifications and viewing angles of the micropillars coated with PDA after laser ablation and selective removal of the PDA from their tips.

**Figure 7 micromachines-13-00371-f007:**
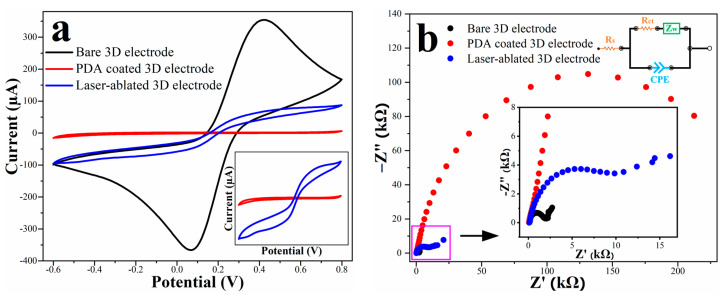
(**a**) Cyclic voltammograms of the bare, PDA-coated and laser-ablated electrodes in PBS solution (10 mM, pH 7.4) containing 10 mM [Fe(CN)_6_]^3−/4−^redox probe at 50 mV s^−1^; (**b**) EIS spectra for the bare, the PDA-coated and laser-ablated electrode and Randles equivalent circuit.

## Data Availability

Data is contained within the article or Appendix A.

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
