# Peer review of "Selective Passivation of Three-Dimensional Carbon Microelectrodes by Polydopamine Electrodeposition and Local Laser Ablation"

_micromachines, 2022, doi:10.3390/mi13030371_

Round 1

Reviewer 1 Report

This manuscript is a work about a selective passivation of microelectrodes by PDA electrodeposition and laser ablation. The authors optimized the electrodeposition to achieve a conformal coating of 3D carbon microelectrodes with PDA completely passivating the electrode surface. The authors removed PDA at the tips of micropillars by laser ablation with high precision, good alignment and good reproducibility. Electrochemical activity of the pillars was approved by scanning electron microscopy, cyclic voltammetry and electrochemical impedance spectroscopy experiments, respectively. I recommend it accepted but requires minor revisions as indicated.

  1. As demonstrated in Figure 1k, the final step is to remove the polydopamine by laser ablation. Also in section 2.5, the authors claimed that the 50ps UV laser (355nm) is with a power 25W at 50%. Although the pattern and position can be well controlled by an xyz machine or an alignment, but it may not easy to control the depth of concentric circles. From Figure 6, we can see a crater-like structures up the tips of micropillars, I think the crater is a side effect of laser ablation. It is suggested to interpretate how the crater is fabricated and its effects on the performance of 3D carbon microelectrodes.
  2. How long does the fabrication take and what about the costs?

Author Response

Reviewer 1

This manuscript is a work about a selective passivation of microelectrodes by PDA electrodeposition and laser ablation. The authors optimized the electrodeposition to achieve a conformal coating of 3D carbon microelectrodes with PDA completely passivating the electrode surface. The authors removed PDA at the tips of micropillars by laser ablation with high precision, good alignment and good reproducibility. Electrochemical activity of the pillars was approved by scanning electron microscopy, cyclic voltammetry and electrochemical impedance spectroscopy experiments, respectively. I recommend it accepted but requires minor revisions as indicated.

  1. As demonstrated in Figure 1k, the final step is to remove the polydopamine by laser ablation. Also in section 2.5, the authors claimed that the 50ps UV laser (355nm) is with a power 25W at 50%. Although the pattern and position can be well controlled by an xyz machine or an alignment, but it may not easy to control the depth of concentric circles. From Figure 6, we can see a crater-like structures up the tips of micropillars, I think the crater is a side effect of laser ablation. It is suggested to interpretate how the crater is fabricated and its effects on the performance of 3D carbon microelectrodes.

RESPONSE: First, we would like to appreciate you taking the time to offer us your valuable comments and insights related to the paper. We found your feedback very constructive. We hope you find these revisions rise to your expectations.

Regarding your first comment about the control on the depth of the laser ablation process. Indeed, the observed crater is a result of laser ablation. We would like to mention that several preliminary experiments were done to optimize the laser power and determine the optimum condition to have uniform removal without damaging the whole structure of the pillar and the passivation layer on the side of the pillars. Higher laser power resulted in destruction of the pillar electrodes, while lower laser power did not allow reproducible complete removal of the PDA layer. Based on your valuable comment the below mentiond sentence is added to the manuscript:

“In preliminary experiments, the effect of laser power was investigated and finally set at 50% of 25 W. Higher laser power induced damage of the 3D carbon electrodes, while lower laser power resulted in incomplete removal of the PDA layer on the top of the pillars.”

However, we would like to mention that the observed crater is not a problem. On the opposite it is advantageous because it increases the electrode surface area at the pillar tip.

Since an identical pattern (concentric circles for laser pathway, as shown in Figure R1) and laser power were used for all the 284 micropillars, very uniform and repeatable results in the depth of the laser-ablated crater on the pillars were observed (Figure R2). 

Figure R1: Schematic illustration of the laser pattern utilized for removing the PDA coating layer from the tip of a micropillar.

Figure R2. SEM images with different tilt angles showing the repeatability and reproducibility of the laser ablation method

Regarding the second part of your comment, you are completely right, the formation of such crater-like structures on the tips of the micropillars is due to the concentric circles in the laser pattern (Figure R1) utilized to remove the PDA coating layer from the tips of the micropillars. For a circular laser pattern, the areal energy density (or dose in J/m2) is higher for the inner edge of the circle than for the outer one, which means that accumulated laser ablation towards the centre of the pillar should be more prominent that towards the edge.   Regarding your invaluable comment and to better clarify the reason for the formation of such crater-like structures (or as we called it concave structures), in the revised version of the manuscript, a schematic of the laser pattern is added to the SI. Furthermore we tried to be more precise in our explanation of the crater-like pattern modifying the text to:

“The formation of concave structures on the tip of the micropillars (Figure 6, top row) could be ascribed to the utilized laser exposure pattern. For a circular laser pattern, the exposure dose is higher for the inner edge of the circle than for the outer one, which means that accumulated laser ablation towards the centre of the pillars should be more prominent that towards the edge.”

In section 3.5. we explained that the higher peak currents measured for the laser-ablated microelectrodes in comparison with the theoretically expected value (derived from Randles-Sevcik equation) might be due to the larger surface area of concave surfaces than the flat tips of the pillars, which can generate a larger electroactive area and thereby higher redox peak current.

  1. How long does the fabrication take and what about the costs?

RESPONSE: Many thanks for this comment. We are not completely sure if the reviewer refers to fabrication of the complete electrodes or only the selective passivation. The fabrication of the electrodes depends on several parameters, like the CMEMS facilities, the diameter of the furnace, etc. For instance, with the facilities available in the DTU Nanolab cleanroom, we can currently fabricate 4 6-inch wafers with around 80 chips of 3D carbon micropillars on each wafer within 1 week. Deposition of the passivation layer and laser ablation were performed on single chips in this study, but potentially both processes could be done on wafer scale. Electrodeposition of PDA on the surface of 3D PyrC electrodes takes around 20-40 minutes (depends on the scan rate employed for electrodeposition). The local laser ablation process takes around 10-20 minutes for each chip, and the most time-consuming part of this step is alignment, which takes around 10-15 minutes. The laser process itself is very short (around 5-7 minutes for all the 284 micropillars on one chip, correspondingly multiplied if wafer scale processing is implemented). The most costly part in the fabrication process of such localized microelectrodes is the cleanroom-based CMEMS process, which is estimated to around 20€ for each chip. Considering the final application of such microdevices to be used as implants for retinal prosthesis or as smart cell substrates for localized cell stimulation, the final price of the product might be less important than the efficiency and biocompatibility of the microdevices. 

We decided not to include specific information about fabrication time and costs in the manuscript, because this information largely depends on the cleanroom settings and equipment available. For the laser ablation process, information about processing time for a single chip was already included in the previous version of the manuscript.

Reviewer 2 Report

The paper entitled "Selective Passivation of Three-Dimensional Carbon Microelectrodes by Polydopamine Electrodeposition and Local Laser Ablation" studies a novel method to fabricate and selectively passivate 3D pyrolytic carbon electrodes composed of micropillars with polydopamine. The scientific quality of the work is high and the scientific contain is complete and clear. One suggestion is to increase the definition of the figures ; the text is unclear because of poor quality. The font of the figures could be also increased.

Reviewer 3 Report

This very well written paper explains almost perfectly what you have done to develop these dopamine coated pyrolized SU-8 pillar electrodes, and I don't have much to add to that (see some comments in the attachment, though...). However, you almost completely miss comparison to earlier work, i.e. how your electrodes and their novel insulator layer compare to other pillar electrodes of ~similar size and other insulator materials. For example having impededance comparison at 1 kHz and maybe some other characteristics and also some evaluation how good insulator polydopamine is compared e.g. with more commonly used SiO2, SiN and SU-8 would be very beneficial additions to your paper. Now everybody knows how to make your electrodes but cannot know whether they are worth to try compared to some other approaches.

Round 2

Reviewer 3 Report

Thank you for your reply. I really wonder why you did not get the file with my other comments, and especially why the editorial office did not contact me if there were some problems. Anyway the commented file is added here again, so please check it and consider making some changes to your manuscript. And what comes to your comments that you were able to make, I still prefer that you include more comparison. If the question is about different dimensions or electrolyte solutions or different impedance measurement methods, I recommend you to fabricate some more samples with more compatible dimensions and also use such measurement methods and electrolyte solutions that are compatible with other publications. Or as you discussed, fabricate samples with some better known insulator. As I mentioned already earlier, it is pointless to go to your concept if one has no idea how good it is compared to others.

Round 3

Reviewer 3 Report

Thank you for your sharp responses. I would not use "only a few" for 10-15 minutes, but never mind...